# Comprehensive Multi-Spectroscopy and Molecular Docking Understanding of Interactions between Fermentation-Stinky Compounds and Mandarin Fish Myofibrillar Proteins

**DOI:** 10.3390/foods12102054

**Published:** 2023-05-19

**Authors:** Jia-Nan Chen, Hui-Lin Zhao, Yu-Ying Zhang, Da-Yong Zhou, Lei Qin, Xu-Hui Huang

**Affiliations:** National Engineering Research Center of Seafood, Collaborative Innovation Center of Seafood Deep Processing, School of Food Science and Technology, Dalian Polytechnic University, Dalian 116034, China; 13591662115@163.com (J.-N.C.); expueryzhl@126.com (H.-L.Z.); zhangyuying99210@163.com (Y.-Y.Z.); zdyzf1@163.com (D.-Y.Z.); qinlei@dlpu.edu.cn (L.Q.)

**Keywords:** mandarin fish myofibrillar protein, interaction, fermentation-stinky, multi-spectroscopy, molecular docking

## Abstract

The release of flavor compounds is a critical factor that influences the quality of fermented foods. A recent study investigated the interactions between four fermentation-stinky compounds (indole, isovaleric acid, dimethyl disulfide, and dibutyl phthalate) and myofibrillar proteins (MPs). The results indicated that all four fermentation-stinky compounds had different degrees of binding to MPs, with dibutyl phthalate and dimethyl disulfide exhibiting stronger interactions. Reduced hydrophobicity enhanced these interactions. Multi-spectroscopy showed that static fluorescence quenching was dominant in the MPs-fermentation-stinky compound complexes. The interaction altered the secondary structure of MPs, predominantly transitioning from β-sheets to α-helix or random coil structures via hydrogen bond interactions. Molecular docking confirmed that these complexes maintained steady states due to stronger hydrogen bonds, van der Waals forces, ionic bonds, conjugate systems, and lower hydrophobicity interactions. Hence, it is a novel sight that the addition of hydrophobic bond-disrupting agents could improve the flavor of fermented foods.

## 1. Introduction

Flavor is one of the key elements influencing the preferences of consumers [1]. Currently, many different types of traditional fermented foods are consumed all over the world, especially in Asia and Europe. Examples include *Chouguiyu* [2,3] and stinky tofu [4] in China, *Surstromming* in Sweden [5], and *Kiviak* in Canada. However, these fermented foods have a shared feature; they give off an unpleasant odor that is unbearable to some people. This limits some people from the delicious pursuit of fermented food. Understanding the fermentation-stinky compounds better is necessary in order to discern how to lessen the stink without altering the characteristics of the whole flavor.

The production of these fermentation-stinky compounds is primarily determined by the following two factors. (1) Under the metabolic reactions of microorganisms, foods undergo a complex set of related reactions during fermentation, involving lipid oxidation, hydrolysis of carbohydrates, proteolysis, etc., which produce low molecular weight stinky compounds [2,6]. These fermentation-stinky compounds include nitrogen and sulfur compounds produced by amino acid hydrolysis and short-chain fatty acids (SCFAs) produced by lipid oxidation [2,7]. For example, indole, isovaleric acid, dimethyl disulfide, and dibutyl phthalate are typical stinky compounds found in fermented mandarin fish [2,3,8]. Obviously, the majority of previous studies have focused on the formation of the fermentation flavor. (2) The release of these key compounds could affect the flavor of fermented foods. Complex matrixes (proteins, lipids, carbohydrates, etc.) interact with flavor compounds in fermented foods, particularly myofibrillar proteins (MPs).

Generally, MPs account for more than 50% of the total protein content in meats, especially over 60% in fish muscle [9]. MPs play an important role in forming meat flavors, since they can bind with the flavor molecules by interaction, such as aldehydes or ketones [10,11,12], and affect the sensory and consumption quality of foods. Flavor molecules can bind to MPs by hydrophobic bonds, ionic bonds, van der Waals forces, covalent bonds, etc. due to differences in their functional groups and binding sites [13]. Moreover, the interaction between flavor molecules and MPs has the potential to alter the secondary structure of MPs. Shen et al. [14] found that the α-helix content gradually decreased with increasing concentrations of ketones. Furthermore, MPs can be utilized as flavor carriers to deliver and preserve the flavors of food components. This property is especially valuable in the creation and formulation of unique food flavors [15]. During food processing, many environmental conditions could affect the binding capacity, such as microwaves, pH, high pressure, oxidation environment, etc. [16]. The reversible binding effect can help minimize flavor loss during food processing and release the flavor components upon consumption, whereas the nonreversible interaction plays a crucial role in removing undesirable off-flavors from foods [17]. Understanding the release mechanism of the fermentation-stinky compounds is significant for improving the flavor of fermented foods. Multi-spectroscopy methods are powerful tools to clarify the changes in the structures of MPs. However, the interactions between fermentation-stinky compounds and MPs have not been fully explored.

The main aims of the current study were to use a typical example of fermented fish to explore the release mechanism of fermentation-stinky compounds through the interactions between them and MPs, including (i) clarifying the binding performance and forces between the typical fermentation-stinky compounds and mandarin fish MPs using GC-MS and various bond-disrupting agents, (ii) depicting the effects of the interactions on the structure of MPs using circular dichroism spectroscopy, fluorescence quenching, and synchronous fluorescence, and (iii) revealing their binding sites by molecular docking. These data could be useful to improve the flavor of fermented fish products.

## 2. Materials and Methods

### 2.1. Materials and Reagents

Live mandarin fish (*Siniperca chuatsi*) was transported from a fish farm in Guangdong and purchased from the Changxing market (Dalian, China). Stinky standards (indole, isovaleric acid, dimethyl disulfide, and dibutyl phthalate) were purchased from Macklin (Shanghai, China). Isotope internal standards (D_26_-dodecane, D_28_-tridecane, D_34_-hexadecane, and D_40_-nonadecane) were purchased from Sigma (Carlsbad, CA, USA). All standards were 99.9% GC.

### 2.2. Preparation of MP Solution

MPs were extracted using the improved method of Xie et al. [18]. Fish were slaughtered and cleaned. Then, the skins, fishbones, and internal organs were removed. The back muscles were cut into bits for preparation in an icebox. The muscles were then chopped into surimi by a meat grinder in an ice bath. The surimi and 0.1 M PBS (pH 7, K_2_HPO_4_/KH_2_PO_4_/KCl buffer) were mixed at 1:4 (*w*/*v*) and homogenized at 8000 rpm at 4 °C for 3 min, and then centrifuged at 8000× *g* at 4 °C for 10 min. The precipitate was duplicated in the above program two times. Next, the precipitate was homogenized with a 0.1 M NaCl solution and homogenized at 8000 rpm at 4 °C for 3 min, and then centrifuged at 8000× *g* at 4 °C for 10 min three times. The precipitate was filtered using 4 layers of gauze and stored at 4 °C for the next experiment (<48 h). Finally, MPs were dissolved in a 0.6 M NaCl solution and diluted into a 5 mg/mL MP solution. The concentration of MPs was detected by a bicinchoninic acid (BCA) kit (Jian Cheng, Inc., Nanjing, China).

### 2.3. Preparation of Fermentation-Stinky Standards and Bond-Disrupting Agent Solutions

Indole, isovaleric acid, dimethyl disulfide, and dibutyl phthalate were diluted into a 1 g/L stock solution by methanol, respectively. The mixture isotope internal standards (10 μg/mL) were prepared by dissolving D_26_-dodecane, D_28_-tridecane, D_34_-hexadecane, and D_40_-nonadecane in n-hexane. The stock solutions of various bond-disrupting agents were prepared, including 2 M Na_2_SO_4_, 10 M urea, 4 M guanidine hydrochloride (GH), and 80% (V/V) propylene glycol (PG).

### 2.4. SPME-GC-MS Analysis

#### 2.4.1. Preparation of Samples for GC-MS

(1)Different concentrations of fermentation-stinky compounds

A total of 5 mL of MP solution, 20 μL of internal standards, and fermentation-stinky standards of different volumes were placed in a 20 mL headspace bottle, and the final fermentation-stinky standards concentration reached 2, 5, and 8 mg/L. MPs and fermentation-stinky standards were vortexed at 2500 rpm/min to bind adequately at 298 K for 12 h. The control samples contained stinky standards and 5 mL of 0.6 M NaCl solution without MPs.

(2)Different bond-disrupting agents

The samples were prepared using the method established by Wang et al. [19], which involved adding 5 mL of MP solution, 20 μL of internal standards, fermentation-stinky standards (the final concentration reached 5 mg/L), and bond-disrupting agents (Na_2_SO_4_, urea, GH, or PG of different volumes were added, and the final concentration reached 0.5 M, 5 M, 1 M, and 20% (V/V), respectively). The control samples contained stinky standards and MP solutions without bond-disrupting agents.

#### 2.4.2. GC-MS Parameters

Free fermentation-stinky compounds were extracted using a PAL auto-injection system equipped with DVB/CAR/PDMS SPME fiber (1 cm, 50/30 μm; 57298-U, Supelco, Bellefonte, PA, USA). The samples were incubated at 60 °C for 20 min and then adsorbed for 40 min.

The analyses of free fermentation-stinky compounds were performed using GC-MS (5890B-5977A; Agilent, Santa Clara, CA, USA) equipped with a non-polar capillary column (HP-5 MS; 30 m × 0.25 mm × 0.25 μm, Agilent) [2]. The SPME was desorbed at 250 °C for 5 min. The heating procedure of the column was started at 35 °C and stayed as such for 3 min, before being heated up from 35 °C to 250 °C at the rate of 5 °C/min, and then the temperature was maintained at 250 °C for 10 min. The carrier gas was helium. The ion source was EI, and the ionization energy was 70 eV. The mass scan range of ion fragments was 35–500 *m*/*z* in full-scan mode. Next, the percentage of free fermentation-stinky compounds was calculated by the following formula.
$$Percentage\;of\;free\;flavor\;compound\left(\%\right)=\frac{A_{MPs}\times k_{2}}{A_{NaCl}\times k_{1}}\times 100\%$$
where *A_NaCl_* and *A_MPs_* represent the peak areas of the NaCl and MP treatment, respectively; NaCl treatment represents only NaCl without MPs and *k*_1_ and *k*_2_ represent the calibration coefficients of NaCl and MPs treatment. The calibration coefficient (*k*) was used to calibrate the peak area and was calculated by the ratio of the peak area of internal standards in the NaCl and MP treatment.

### 2.5. Fluorescence Analysis

#### 2.5.1. Preparation of Samples for Fluorescence Analysis

Free fermentation-stinky standards were added to 1 mL (5 mg/mL) of MP solution, reaching a final concentration of 0, 2, 5, or 8 mg/L. For the fluorescence quenching analysis, samples were vortexed at 2500 rpm/min at 298 K for 40 min. For the thermodynamic stability analysis, samples were mixed thoroughly and then treated in a water bath at different temperatures (293 K, 303 K, or 313 K) for 40 min.

#### 2.5.2. Fluorescence Quenching Spectra

Fluorescence quenching was determined by an F-2700 fluorescence spectrometer (Hitachi, Tokyo, Japan) at a wavelength of 280 nm. Both the slits of E_x_ and E_m_ were 10 µm. The FWT voltage was 400 V. The interactions between fermentation-stinky compounds and MPs were assessed through the Stern–Volmer equation [20].
$$\frac{F_{0}}{F}=1+K_{q}\tau_{0}\left[Q\right]=1+K_{SV}\left[Q\right]$$
where *F* and *F*_0_ represent the fluorescence intensity with and without quencher, respectively; [*Q*] represents the concentration of quencher; *K_q_* represents the bimolecular quenching constant; *τ*_0_ is the average lifetime of fluorescence molecules without quencher (10^−8^); *K_sv_* is the quenching rate constant. The Stern–Volmer equation can be modified to calculate *n* (the number of binding sites) and *K_a_* (the equilibrium quenching constant, which is used to judge the type of quenching) as follows:$$\log(\frac{F_{0}-F}{F})=\log K_{a}+n\log\left[Q\right]$$

#### 2.5.3. Thermodynamic Stability of the MPs-Fermentation-Stinky Compound Complexes

Thermodynamic stability is a factor to assess the interaction mode between MPs and fermentation-stinky compounds at different temperatures, especially the enthalpy change (Δ*H*) and entropy change (Δ*S*), which were calculated by the Van’t Hoff equation [21].
$$lnK=-\frac{∆H}{RT}+\frac{∆S}{R}$$
where *R* is the gas constant and the value is 8.314 J·mol^−1^·K^−1^; *K* represents the effective quenching constant (*K_a_*) at the corresponding temperature (*T*). Therefore, the free energy (Δ*G*) could be calculated by the following:∆G=−RTlnK=∆H−T∆S

#### 2.5.4. Synchronous Fluorescence Spectra

A fixed excitation wavelength (Δλ = λ_Em_ − λ_Ex_) of 15 or 60 nm was used to analyze the synchronous fluorescence spectra. Other parameters which were the same were used for fluorescence quenching analysis.

### 2.6. Circular Dichroism (CD) Spectra

The samples were prepared with the same treatment as the above fluorescence samples. In particular, the 8 mg/L samples were used to render the changes in the relative percentage of the secondary structure. The secondary structure (α-helix, β-sheet, β-turns, and random coil) of MPs-fermentation-stinky compound complexes was determined using the J-1500 CD spectrometer (JASCO, Tokyo, Japan). The scan range was 200–260 nm at 297 K. High-purity nitrogen was purged from gas at a rate of 5 L/min. The HT voltage was 300–600 V. The secondary structures of α-helix, β-sheet, β-turns, and random coils of MPs-fermentation-stinky compound complexes were calculated using Yang’s method [22].

### 2.7. Molecular Docking

Firstly, the sequence of receptor MP derived from mandarin fish was matched using the UniProtKB (www.uniprot.org) accessed on 5 January 2023. Secondly, the homologous receptor protein that assessed the high scores of similarity to the sequence was obtained from the RCSB PDB (www.rcsb.org) accessed on 5 January 2023. Thirdly, ChemDraw software was exploited to create the small molecule ligands. Finally, MD was performed using Discovery Studio 2019 in LibDock mode.

### 2.8. Statistical Analysis

Analysis of the significance (*p* < 0.05) of different treatments and variance was accomplished using DPS 7.5. The visualization was performed using GraphPad Prism 8.

## 3. Results and Discussion

### 3.1. The Interactions between MPs and Fermentation-Stinky Compounds

Previous studies showed that indole, isovaleric acid, dimethyl disulfide, and dibutyl phthalate were identified as the key stinky compounds in fermented mandarin fish [2,7,8]. These four stinky compounds were used to explore the interactions between MPs and fermentation-stinky compounds, whose structures are shown in Figure 1. Figure 2A observed the percentage of free fermentation-stinky compounds at different concentrations. The percentage of free fermentation-stinky compound solution (without MPs) was set to 100% (Figure 2). If the percentage is less than 100%, the interaction between the compound and MPs occurs. Lower free percentages of fermentation-stinky compounds reflected stronger binding affinities. The binding affinities of all the free fermentation-stinky compounds significantly (*p* < 0.05) increased as the additional concentration increased from 2 to 8 mg/L, which could be attributed to the presence of more binding sites [19]. Binding to MPs increased first, then decreased as isovaleric acid and dimethyl disulfide concentrations significantly increased (*p* < 0.05). Yet both the free percentages of isovaleric acid and dimethyl disulfide reached a low at 5 mg/L, indicating the binding sites had been saturated on MPs. In particular, dimethyl disulfide could bind to higher compound concentrations and present better binding capacity compared to other compounds. Whereas dibutyl phthalate and indole had the strongest interactions with MPs at 2 mg/L, their binding capacity was gradually reduced, reaching site saturation. Dibutyl phthalate and MPs could occur through the action of hydrogen bonds and van der Waals. Moreover, hydrophobic interactions between small molecules and proteins can also affect the binding capacity of MPs [10]. The order of binding capacity to MPs at the optimal concentration was dibutyl phthalate = dimethyl disulfide > indole = isovaleric acid (*p* < 0.05), with a percentage of 85.7%, 82.8%, 51.1%, and 49.1%, respectively.

### 3.2. Effects of Bond-Disrupting Agents on the Binding Capacity

The interactions between MPs and fermentation-stinky compounds could be attributed to the actions of various forces. Figure 2B displays the actions of different bond-disrupting agents. Na_2_SO_4_ is regarded as a salt that stabilizes proteins, which can enhance their hydrophobicity through intra- and intermolecular interaction [23]. Figure 2C observed that the binding of indole, isovaleric acid, dimethyl disulfide, and dibutyl phthalate to MPs was significantly reduced (*p* < 0.05). Contradictory results were reported for aldehydes and furans [24,25]. The increased surface hydrophobicity of MPs limited their binding to fermentation-stinky compounds. This may be related to the hydrophobicity of fermentation-stinky compounds themselves. The partition coefficient-P (log P) value is used to assess the hydrophobicity of the compound, and a high log P indicates that the compound has a strong hydrophobicity [26]. As shown in Figure 1, all the log *p* values of these fermentation-stinky compounds were greater than one with strong hydrophobicity, especially dibutyl phthalate, which reached 4.82. Therefore, this explained the highest presence of the free percentage of dibutyl phthalate (Figure 2C).

Urea can denature proteins by decreasing their hydrophobicity and destroying hydrogen bonds [11]. The importance of hydrophobic interactions and hydrogen bonds for MPs-fermentation-stinky compound systems was thus highlighted by observing the changes in binding capacities in the presence of urea (Figure 2C). In contrast to urea, GH is a protein denaturant that destroys ionic bonds in addition to hydrophobic and hydrogen bonds [27]. Significant reductions in the binding of MPs to dimethyl disulfide were observed in the GH sample compared to the urea treatment. This indicated that hydrogen bond interactions provided higher contributions to the MPs-dimethyl disulfide system than hydrophobic interactions. Yet hydrogen bond interactions did not significantly affect the binding of indole or dibutyl phthalate to MPs. Moreover, PG inhibits hydrophobic interactions while promoting hydrogen and ionic bonds [28]. For dimethyl disulfide, the presence of PG in the binding capacity was between urea and GH, in which ionic bone interaction occurred. Compared to the GH samples, the PG samples in indole, isovaleric acid, and dibutyl phthalate presented lower interactions in the MPs-fermentation-stinky compound systems. In brief, hydrogen and ionic bonds could compete with electrostatic or other forces for the binding sites of MPs.

### 3.3. Fluorescence Quenching of MPs by Fermentation-Stinky Compounds

Fluorescence quenching reflects changes near fluorescent groups in proteins. Generally, Tyr and Trp can exhibit autofluorescence with an excitation wavelength of 280 nm [29]. As shown in Figure 3A–D, a characteristic fluorescence peak of Tyr and Trp around 340 nm was observed in the MP solution, the intensity of which decreased as the concentration of fermentation-stinky compounds increased. In particular, the fluorescence quenching from indole was strongest (Figure 3A). This indicated that these four fermentation-stinky compounds interacted with MPs, exposing Tyr or Trp residues, which could result in changes in the secondary and tertiary structures of MPs.

In order to explain the quenching mechanism, the Stern Volmer parameters, such as the bimolecular quenching constant (*K_q_*) and the quenching rate constant (*K_sv_*), are shown in Table 1. Static quenching is indicated by a decrease in *K_sv_* with increasing temperature [27]. The *K_sv_* values of dimethyl disulfide and dibutyl phthalate increased from 293 K to 313 K, where dynamic quenching was dominant. At 303 K, the *K_sv_* values of indole and isovaleric acid were the highest or lowest. This indicated that MPs exhibited both dynamic and static quenching. Furthermore, the *K_q_* values of all these fermentation-stinky compounds were much greater than the maximum of the dynamic quenching rate constant (2 × 10^10^ M^−1^ S^−1^), indicating that static quenching also contributes to the interactions. The equilibrium constants (*K_a_*) were greater than 10^3^ M^−1^, which revealed the strong interactions.

### 3.4. Thermodynamic Stability Analysis

The thermodynamic stability of these MPs-fermentation-stinky compound complexes was reflected in the change in enthalpy change (Δ*H*) and entropy change (Δ*S*), which were assessed by the principle as follows [21]: (1) if Δ*H* > 0 and Δ*S* > 0, hydrophobic interaction is the major interaction; (2) if Δ*H* < 0 and Δ*S* < 0, the van der Waals force or hydrogen bond plays an important role; (3) if Δ*H* < 0 and Δ*S* > 0, the electrostatic force is dominant. The thermodynamic data are shown in Table 1. Consequently, the binding reactions of these four fermentation-stinky compounds and MPs were spontaneous (Δ*G* < 0). Δ*H* < 0 suggested that electrostatic interactions occurred in the binding system [27]. Furthermore, the negative values of both Δ*H* and Δ*S* revealed that van der Waals forces and hydrogen bonding were dominant in the system.

### 3.5. Synchronous Fluorescence Spectra Analysis

Figure 4A–H observed the fluorescence of MPs-fermentation-stinky compound complexes by Δλ = 15 and Δλ = 60, respectively. The fluorescence quenching abilities decreased as the concentration of the additional fermentation-stinky compounds increased. Furthermore, blue shift presented in all fluorescence spectra figures. The binding sites could be located at around Trp and Tyr residues. In brief, Trp and Tyr residues were exposed to a more hydropic environment, the polarity of complexes was reduced, and the hydropic interaction was enhanced. Due to the flexible structure of MP, a conjugate system might exist around the binding sites.

### 3.6. Change in the Secondary Structure of MPs

CD spectra were used to present changes in the secondary structure of MPs-fermentation-stinky compound complexes. As shown in Figure 5A–D, all MPs-fermentation-stinky compound complexes displayed two negative peaks with 208 nm and 222 nm, which illustrated that the n–π* of peptide bonds in an α-helical structure transformed [30]. As the concentration of fermentation-stinky compounds increased, the α-helical structure changed dramatically. Compared to the effects of different fermentation-stinky compounds on the secondary structure of MPs, the abundance of the negative peaks of isovaleric acid (Figure 5B) and dimethyl disulfide (Figure 5C) presented a trend change with concentration. The abundance of the negative peak at 5 mg/L for indole and dibutyl phthalate was comparable to that at 8 mg/L.

Under the addition of 8 mg/L fermentation-stinky compounds, Figure 5E observed that α-helix content was enhanced, with the most significant increase appearing as a result of indole treatment. Their β-sheets were completely transformed, while their random coil content was increased. The more secondary structures of MPs randomly coiled by isovaleric acid or dimethyl disulfide binding, the more their content increased from 39.47% to 48.00% and 48.40%, respectively. The results suggested that interactions between all fermentation-stinky compounds and MPs altered the initial secondary structure of MPs, allowing the β-sheet to fully expand and add α-helix and random coils. Hydrophobic bonding promotes β-sheet formation, while hydrogen bonding dominates the helix [10]. Thus, this indicated that the transition from the β-sheets to the α-helix or random coil in the structure of MPs was dominated by hydrogen bond interactions.

The correlation heatmap (Figure 6) showed that the adsorption capacity of fermentation-stinky compounds and α-helix percent presented a positive correlation, whereas the percentage of β-sheet, β-turns, and random coil and fluorescence intensity at 280 nm, Δλ = 15 nm, and Δλ = 60 nm were negatively related to them. Differently to this, random coil percent and the adsorption capacity of dimethyl disulfide presented a high positive correlation (r = 1.0). Therefore, the change in the secondary structure of MPs and the reduction in fluorescence intensity could be attributed to the binding between fermentation-stinky compounds and MPs.

### 3.7. Analysis of the Binding Sites Based on Molecular Docking (MD)

In mandarin fish MPs, myosin heavy chain-7 was the major protein, and homologized 5TBY (PDB ID) was used as a receptor for MD. A total of 84 active sites were found in 5TBY. The best docking sites were identified by the highest Libdock scores. The 3D and 2D structures of these myosin-fermentation-stinky complexes are shown in Figure 7. The result showed that in the myosin-indole complex, indole bound to Tyr 287, Leu 290, Met 357, and Ile 313 residues by the conjugate system (pi-sulfur and pi-sigma), and Tyr 310, Phe 312, etc. residues by van der Waals force (Figure 7A). In the myosin-isovaleric acid complex, isovaleric acid, Asp 309, and Tyr 310 residues had conventional hydrogen bond interactions, and Pro 307, Ile313, Tyr 308, and Leu 290 residues had conjugate system interactions (Figure 7B). Figure 7C displayed that Tyr 287 residue was the main contributor to the formation of hydrogen bond interactions with dimethyl disulfide, and the corresponding distance was 2.73 Å. Dibutyl phthalate was bound to MPs by various interactions, especially hydrogen bonds interacted with Tyr 310 and Asp 309 residues (Figure 7D).

Gaussian contact maps (Figure 8) observed that hydrogen bonds on the surface of the complexes had either positive or negative effects on the binding system of MPs. The stronger the surface hydrophobicity the complexes presented, the lower the internal hydrophobicity they had. Moreover, the low interpolated charge played a part in the interactions. The MD results could support the above interactions analysis of GC-MS, fluorescence spectra, and CD spectra. To sum up, these four fermentation-stinky compounds presented similar interactions to MPs, which included stronger hydrogen bond, van der Waals force, ionic bone, conjugate system, and lower hydrophobicity interactions.

## 4. Conclusions

This study revealed the release mechanism of four characteristic fermentation-stinky compounds through the interactions between them and MPs. Hydrophobic bond interactions inhibited their binding. Hydrogen bond and van der Waals force interactions maintained the stability of the complexes. The blue shift in synchronous fluorescence spectra confirmed the binding of fermentation-stinky compounds and MPs around the Tyr and Trp residues, which expanded the β-sheet structure. The MD results could support the above interaction analysis of GC-MS, fluorescence spectra, and CD spectra. To sum up, these fermentation-stinky compounds presented similar interactions to MPs, which included stronger hydrogen bonds, van der Waals forces, ionic bones, conjugate systems, and lower hydrophobicity interactions. A novel discovery was the addition of hydrophobic bond-disrupting agents, which can significantly reduce the release of stinky compounds to meet the needs of more consumers, and which was a potential method to control the fermentation-stinky compounds of fermented foods. The overall appearance of fermentation-stinky compounds depends not only on their content, but also on their odor thresholds, interactions with other aroma compounds, etc. However, reducing the odor of fermented food while maintaining its original characteristic flavor needs further study.

## Figures and Tables

**Figure 1 foods-12-02054-f001:**
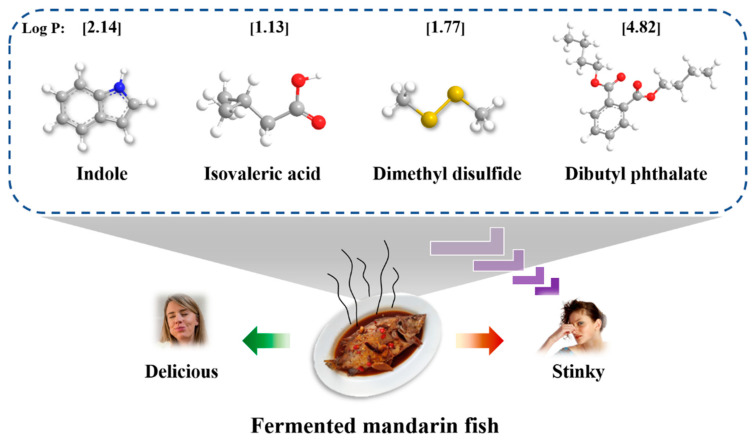
The 3D structures and partition coefficient-P (log P) values of the fermentation-stinky compounds (indole, isovaleric acid, dimethyl disulfide, and dibutyl phthalate) were identified from fermented foods, such as fermented mandarin fish.

**Figure 2 foods-12-02054-f002:**
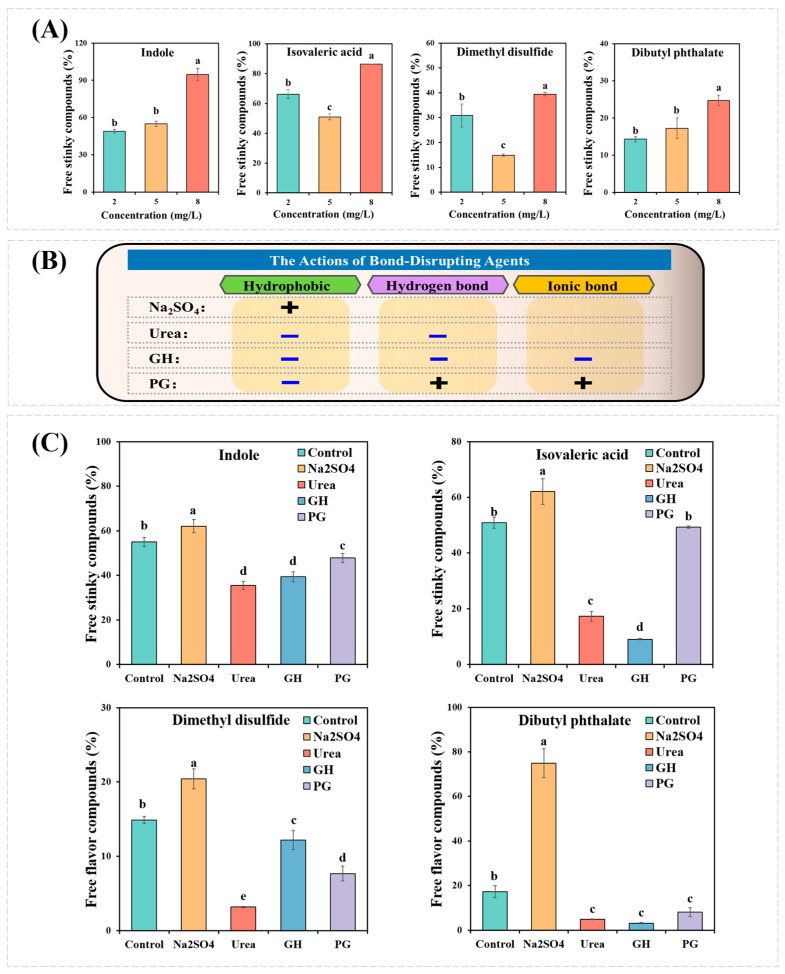
Effects of (**A**) different concentrations and (**C**) different bond-disrupting agents on the free fermentation-stinky compounds. (**B**) The actions of different bond-disrupting agents. Different letters indicate significant differences between groups (*p* < 0.05).

**Figure 3 foods-12-02054-f003:**
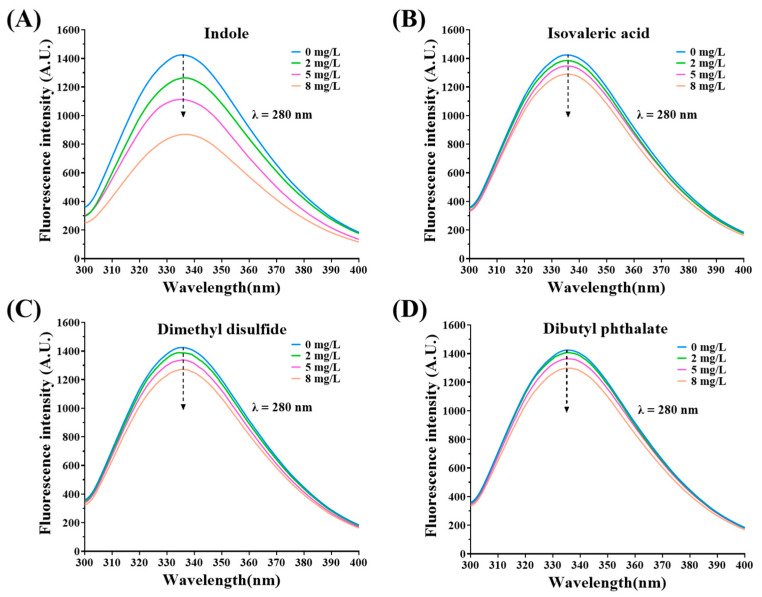
(**A**–**D**) Changes in the fluorescence intensity of MPs-fermentation flavor compounds complexes at λ = 280 nm in different concentrations (0, 2, 5, and 8 mg/L).

**Figure 4 foods-12-02054-f004:**
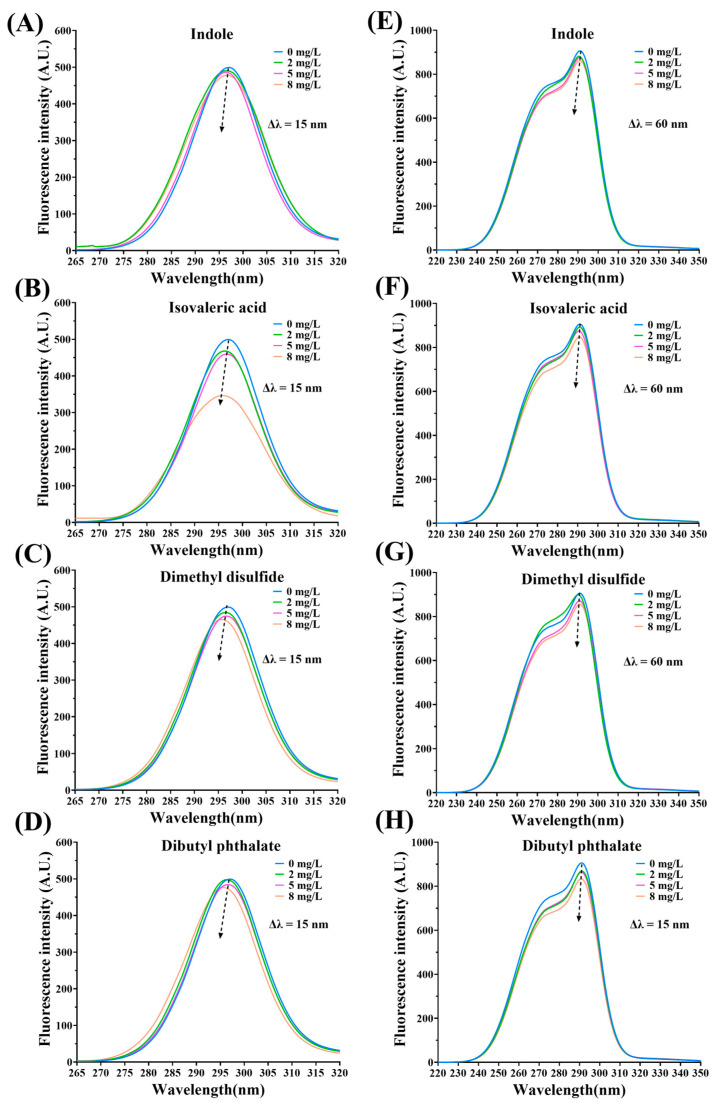
Synchronous fluorescence spectra of fermentation-stinky compounds-MPs complexes at (**A**–**D**) Δλ = 15 nm and (**E**–**H**) Δλ = 60 nm in different concentrations (0, 2, 5, and 8 mg/L).

**Figure 5 foods-12-02054-f005:**
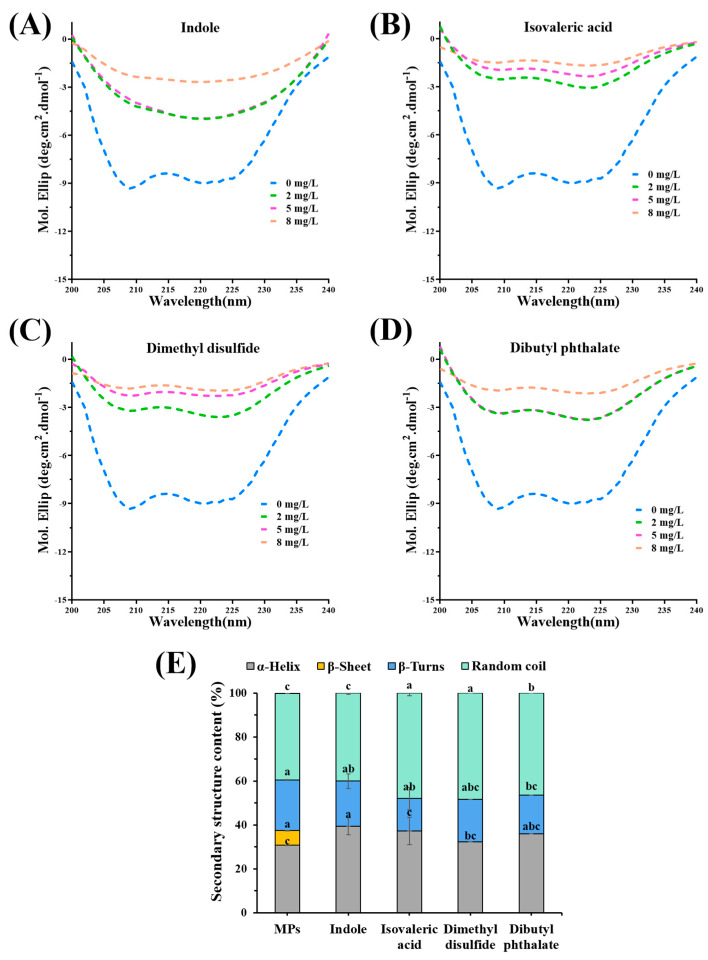
Changes in the circular dichroism spectra of (**A**) myosin-indole complexes, (**B**) myosin-isovaleric acid complexes, (**C**) myosin-dimethyl disulfide complexes, and (**D**) myosin-dibutyl phthalate complexes in different concentrations (0, 2, 5, and 8 mg/L). (**E**) Changes in the secondary structure content of mandarin fish MPs. Different letters indicate significant differences between groups (*p* < 0.05).

**Figure 6 foods-12-02054-f006:**
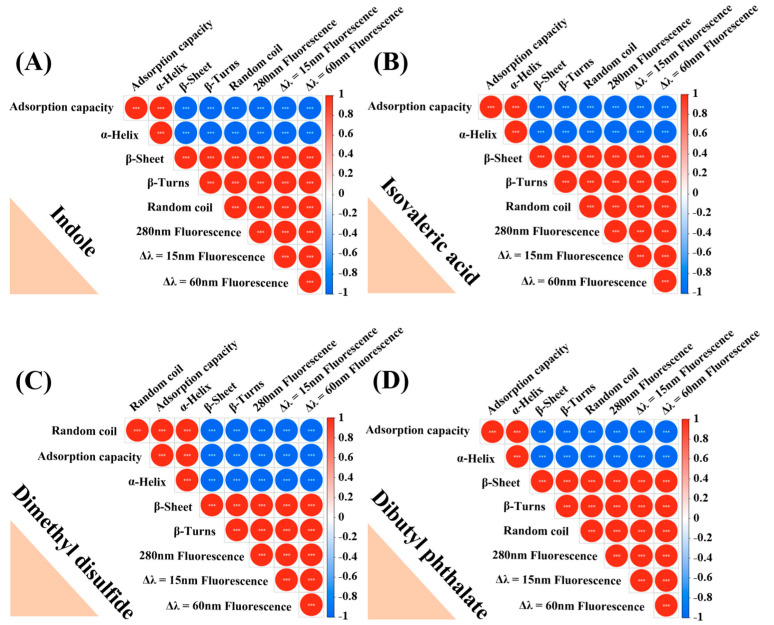
Heatmap of the correlation analysis among the percentage of α-helix, β-sheet, β-turns, and random coil and the fluorescence intensity at 280 nm, Δλ = 15 nm, and Δλ = 60 nm, and the adsorption capacity of (**A**) indole, (**B**) isovaleric acid, (**C**) dimethyl disulfide, and (**D**) dibutyl phthalate. The adsorption capacity was calculated by 100% minus the free stinky compounds (%).

**Figure 7 foods-12-02054-f007:**
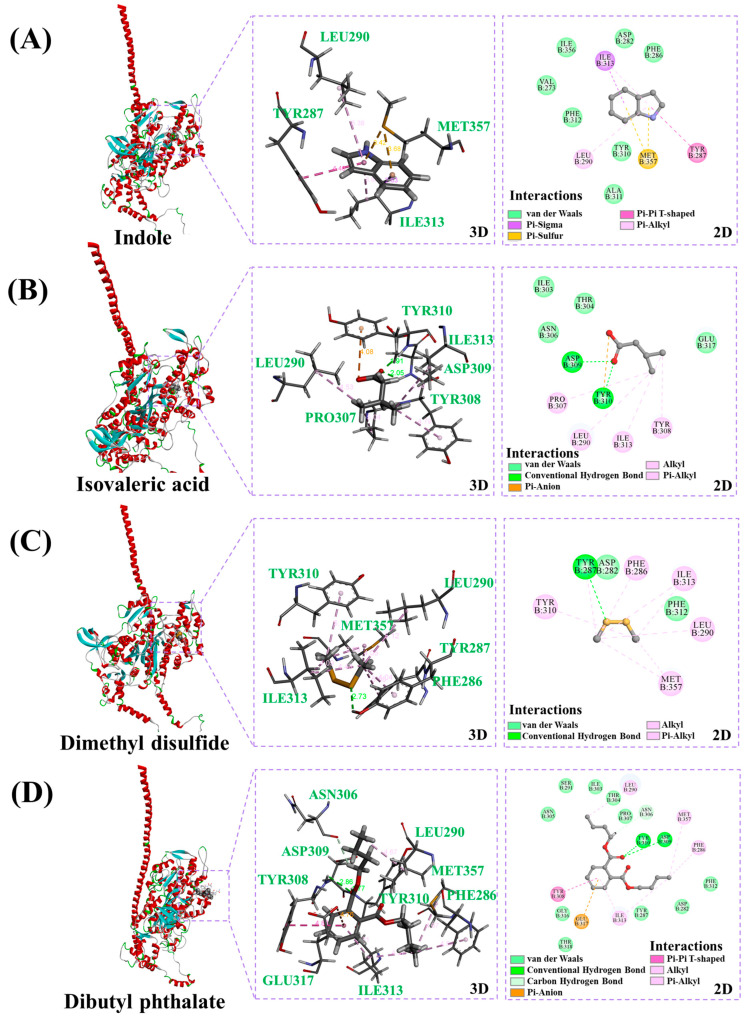
The best docking model, 3D, and 2D structure of (**A**) myosin-indole complexes, (**B**) myosin-isovaleric acid complexes, (**C**) myosin-dimethyl disulfide complexes, and (**D**) myosin-dibutyl phthalate complexes.

**Figure 8 foods-12-02054-f008:**
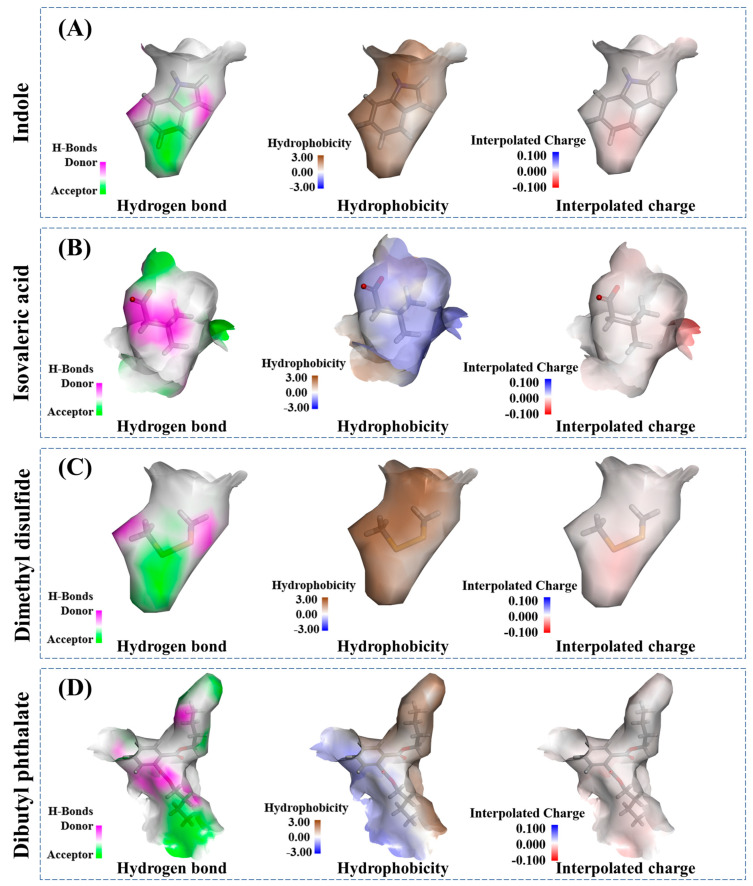
The binding bond distribution of (**A**) myosin-indole complexes, (**B**) myosin-isovaleric acid complexes, (**C**) myosin-dimethyl disulfide complexes, and (**D**) myosin-dibutyl phthalate complexes.

**Table 1 foods-12-02054-t001:** The equilibrium constant *K_a_* and thermodynamic parameters of the complexes between myofibrillar proteins and fermentation-stinky compounds calculated by the Stern–Volmer equation.

Compound	T (K)	*K_sv_* (10^3^M^−1^)	*K_q_* (10^11^M^−1^s^−1^)	*K_a_* (10^4^M^−1^)	*n*	Δ*H* (kJ·mol^−1^)	Δ*S* (J·mol^−1^ K^−1^)	Δ*G* (kJ·mol^−1^)
Indole	293	9.6	9.6	2.57 × 10³	1.1077	−512.42	−1607.01	−41.57
	303	1.8	1.8	3.51	0.5650			−25.50
	313	2.2	2.2	2.05	0.4609			−24.18
Isovaleric acid	293	1.1	1.1	6.18	0.6915	−79.89	−257.51	−4.44
	303	1.4	1.4	2.15	0.5010			−1.86
	313	0.4	0.4	3.06 × 10^8^	2.4453			−47.60
Dimethyl disulfide	293	0.9	0.9	6.30	0.9650	−275.50	−905.81	−10.10
	303	1.2	1.2	1.53	0.4711			−1.04
	313	1.2	1.2	1.60	0.4555			−1.15
Dibutyl phthalate	293	3.8	3.8	1.02 × 10^5^	1.2105	−390.19	−1235.80	−28.10
	303	9.9	9.9	6.40 × 10^2^	0.9438			−15.74
	313	16.1	16.1	1.36 × 10^4^	5.1350			−23.19

*K_a_*, *K_sv_*, and *K_q_* are the equilibrium constant, the quenching rate constant, and the bimolecular quenching constant in the Stern–Volmer equation, respectively. The thermodynamic parameters include enthalpy change (Δ*H*), entropy change (Δ*S*), and free energy change (Δ*G*).

## Data Availability

Data is contained within the article.

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
