# Peer review of "Comprehensive Multi-Spectroscopy and Molecular Docking Understanding of Interactions between Fermentation-Stinky Compounds and Mandarin Fish Myofibrillar Proteins"

_foods, 2023, doi:10.3390/foods12102054_

Round 1
Reviewer 1 Report
This article dressed the interaction between 4 stinky compounds and MP. The data was solid, but the English need to be reviewed. The following are some suggestions.
1. Introduction line 27-29 is not smooth. Line 32-33 is not related to the phrase.
2. Line 51, MPs play an important role in forming meat flavors. Please provide a rederence or explain in detail.
3. Line 55-58 is very hard to read. please consider to reword it.
4. Line 55-56. “The binding of ketones and chicken MPs could 55 break down the α-helix structure”There is no evidence to say alpha-helix structure was break down by ketones. Actually, in reference, it said The measured α-helix “content” decreased gradually with the increase of ketones concentration.
5. Line 84, homogenized at 8000 “r”, do you mean rpm?
6. Line 96 the unit ug/mL please use “μ” symbol
7. Section 2.3, there are only 4 compounds studied in this study. Why 4 isotopically labeled internal standard (D26-dodecane, D28-tridecane, D34-hexadecane, and D40- 96 nonadecane) were used, but they are all hydrocarbons.
8. Line 116, please specify the characteristic of SPME fiber, for example the length, the coating thickness.
9. Line 118 is hard to read. “Free fermentation stinky compounds were performed by the method of Chen, Zhang, Huang, Dong, Dong, Zhou, Zhu and Qin [4] using GC-MS” compounds were performed? What do you mean?
10. Line 120, the column I.D. = 0.25cm ? Please check it, it may be 0.25 mm.
11. Line 124, eV is energy unit not voltage. Please check.
12. Line 125, all-scan mode? Do you mean full-scan mode?
13. Line 201, Fig.2 is the free compound concentration, how does this come with the conclusion of binding affinities increased? Please explain and point out where the data can say in Fig. 2.
There are many grammer errors, please consider make the manuscript to be reviewed by a native speaker.
Author Response
Response to Reviewer 1 Comments
Point 1: Introduction line 27-29 is not smooth. Line 32-33 is not related to the phrase.
Response 1: Thanks for your kind suggestion. Lines 27-29 and 32-33. These parts were deleted.
Point 2: Line 51, MPs play an important role in forming meat flavors. Please provide a reference or explain in detail.
Response 2: Thanks for your kind suggestion. Lines 48-54. These sentences were rewritten as “MPs play an important role in forming meat flavors, which could bind with the flavor molecules by interaction, such as aldehydes or ketones [10-12], and affect the sensory and consumption quality of foods. Flavor molecules can bind to MPs by hydrophobic bonds, ionic bonds, van der Waals forces, covalent bonds, etc. due to differences in their functional groups and binding sites [13]. Besides, the interaction between flavor molecules and MPs has the potential to alter the secondary structure of MPs.”
Point 3: Line 55-58 is very hard to read. please consider to reword it.
Response 3: Thanks for your kind suggestion. Lines 55-62. These sentences were rewritten as “Furthermore, MPs can be utilized as flavor carriers to deliver and preserve the flavors of food components. This property is especially valuable in the creation and formulation of unique food flavors [15]. During food processing, many environmental conditions could affect the binding capacity, such as microwave, pH, high pressure, oxidation environment, etc. [16]. The reversible binding effect can help minimize flavor loss during food processing and release the flavor components upon consumption, whereas the nonreversible interaction plays a crucial role in removing undesirable off-flavors from foods [17].”
Point 4: Line 55-56. “The binding of ketones and chicken MPs could 55 break down the α-helix structure” There is no evidence to say alpha-helix structure was break down by ketones. Actually, in reference, it said The measured α-helix “content” decreased gradually with the increase of ketones concentration.
Response 4: Thanks for your kind suggestion. Lines 52-55. These sentences were rewritten as “Besides, the interaction between flavor molecules and MPs has the potential to alter the secondary structure of MPs. Shen, et al. [14] found that the α-helix content gradually decreased with increasing concentrations of ketones.”
Point 5: Line 84, homogenized at 8000 “r”, do you mean rpm?
Response 5: Thanks for your kind suggestion. Lines 88, 91. “8000 r” was revised as “8000 rpm”.
Point 6: Line 96 the unit ug/mL please use “μ” symbol
Response 6: Thanks for your kind suggestion. Line 99. “ug/mL” was revised as “μg/mL”.
Point 7: Section 2.3, there are only 4 compounds studied in this study. Why 4 isotopically labeled internal standard (D26-dodecane, D28-tridecane, D34-hexadecane, and D40-nonadecane) were used, but they are all hydrocarbons.
Response 7: Thanks for your comment. Lines 133-137. MPs can produce volatile compounds, which could interfere with the accuracy of the results. Generally, isotope compounds don’t exist naturally in MPs and have greater stability. The addition of isotopes can better observe the signal intensity of the sample on the instrument. Better isotope peak area reproduction shows more accurate results and reduces these errors on the instrument. These isotopes of these compounds are a better choice for the precise calculation of the content. However, isotope flavor compounds are too expensive, and we replace them with isotope n-alkanes in our laboratory, which are acceptable only for the calibration of the peak area. Therefore, isotope internal standards were used to calibrate the peak area of the stinky compounds.
Point 8: Line 116, please specify the characteristic of SPME fiber, for example the length, the coating thickness.
Response 8: Thanks for your kind suggestion. Lines 121. Detailed information on DVB/CAR/PDMS SPME fiber (1 cm, 50/30 μm; 57298-U, Supelco, Bellefonte, PA, USA) was revised.
Point 9: Line 118 is hard to read. “Free fermentation stinky compounds were performed by the method of Chen, Zhang, Huang, Dong, Dong, Zhou, Zhu and Qin [4] using GC-MS” compounds were performed? What do you mean?
Response 9: Thanks for your comment. Lines 124-126. We want to cite the previous method. This part was revised as “The analyses of free fermentation-stinky compounds were performed using GC-MS (5890B-5977A; Agilent, California, USA) equipped with a non-polar capillary column (HP-5 MS; 30 m × 0.25 mm × 0.25 μm, Agilent) [2].”
Point 10: Line 120, the column I.D. = 0.25 cm ? Please check it, it may be 0.25 mm.
Response 10: Thanks for your comment. Line 126. We want to cite the previous method. “0.25 cm” was revised as “0.25 mm”.
Point 11: Line 124, eV is energy unit not voltage. Please check.
Response 11: Thanks for your kind suggestion. Line 129. “the ionization voltage” was revised as “the ionization energy”.
Point 12: Line 125, all-scan mode? Do you mean full-scan mode?
Response 12: Thanks for your kind suggestion. Line 130. “all-scan mode” was revised as “full-scan mode”.
Point 13: Line 201, Fig.2 is the free compound concentration, how does this come with the conclusion of binding affinities increased? Please explain and point out where the data can say in Fig. 2.
Response 13: Thanks for your comment. Lines 200-203. “The percentage of free fermentation-stinky compound solution (without MPs) was set to 100% (Figure. 2). If the percentage is less than 100%, the interaction between the compound and MPs occurs. Lower free percentages of fermentation-stinky compounds reflected stronger binding affinities.” was added.
Point 14: This article dressed the interaction between 4 stinky compounds and MP. The data was solid, but the English need to be reviewed.
Response 13: Thanks very much for your valuable and professional comments. The manuscript has been revised and re-polished by a native English speaker.

Reviewer 2 Report
Dear Authors,
In general, the manuscript is good to read, the structure of the work is clear. After reading the paper, it is clear that the authors performed a lot of experiments and analyzes in order to obtain detailed research results. I present my comments below:
1. Introduction. Lines 27-28. This sentence is incomprehensible in the context of the entire thread. Please correct.
2. Line 74. “Mandarin fish was collected from the Changxing market”. - There is not enough information about the fish.
3. Line 116. Why did the Authors choose such a fibre?
4. All manuscript. For example. Line 87, 88: The space after 4 degrees Celsius is unnecessary. Line 83 is: “0.1M PBS”; should be: “0.1 M PBS”?
5. Figure 3. Sygnatury osi są bardzo małe.
6. Basic statistical methods are used. It is a pity that the authors did not use an in-depth statistical method to show the correlation.
7. The literature review is almost sufficient, although 26 items is not a very extensive literature review.
Author Response
Response to Reviewer 2 Comments
Point 1: Introduction. Lines 27-28. This sentence is incomprehensible in the context of the entire thread. Please correct.
Response 1: Thanks for your kind suggestion. Lines 27-28. This sentence was deleted.
Point 2: Line 74. “Mandarin fish was collected from the Changxing market”. - There is not enough information about the fish.
Response 2: Thanks for your kind suggestion. Lines 77-78. “Live mandarin fish (Siniperca chuatsi) was transported from a fish farm in Guangdong and purchased from the Changxing market (Dalian, China).” was added.
Point 3: Line 116. Why did the Authors choose such a fibre?
Response 3: Thanks for your comment. Line 121. The fiber be filled in divinylbenzene/carboxen/polydimethylsiloxane (DVB/CAR/PDMS). This fiber can adsorb C3-C20 (about MW 40-275) compounds, and the adsorption range and polarity are suitable for aroma compound analysis. Besides, there are some types such as polydimethylsiloxane (PDMS, MW 125-600), carboxen/polydimethylsiloxane (CAR/PDMS, MW 30-225), and polyacrylate (PA, MW 80-300). Thus, DVB/CAR/PDMS was used.
Point 4: All manuscript. For example. Line 87, 88: The space after 4 degrees Celsius is unnecessary. Line 83 is: “0.1M PBS”; should be: “0.1 M PBS”?
Response 4: Thanks for your kind suggestion. Lines 122, 126-128. The spaces after 4 degrees Celsius were deleted. Lines 87 and 110. The space issues were corrected. The “0.1M PBS” was revised as “0.1 M PBS”.
Point 5: Figure 3. Sygnatury osi są bardzo małe.
Response 5: Thanks for your kind suggestion. Lines 254 and 301. Figure 3 and Figure 4 were improved.
Point 6: Basic statistical methods are used. It is a pity that the authors did not use an in-depth statistical method to show the correlation.
Response 6: Thanks for your kind suggestion. Lines 337-344. “The correlation heatmap (Figure. 6) showed that the adsorption capacity of fermentation-stinky compounds and α-helix percent presented a positive correlation, whereas the percentage of β-sheet, β-turns, and random coil and the fluorescence intensity at 280 nm, Δλ = 15 nm, and Δλ = 60 nm were negatively related to them. Differently, random coil percent and the adsorption capacity of dimethyl disulfide presented a high positive correlation (r = 1.0). Therefore, the change in the secondary structure of MPs and the reduction in fluorescence intensity could be attributed to the binding between fermentation-stinky compounds and MPs.” were added.
Figure 6. Heatmap of the correlation analysis among the percentage of α-helix, β-sheet, β-turns, and random coil and the fluorescence intensity at 280 nm, Δλ = 15 nm, and Δλ = 60 nm, and the adsorption capacity of (A) indole, (B) isovaleric acid, (C) dimethyl disulfide, and (D) dibutyl phthalate. The adsorption capacity was calculated by 100% minus the free stinky compounds (%).
Point 7: The literature review is almost sufficient, although 26 items is not a very extensive literature review.
Response 7: Thanks for your kind suggestion. We supplied some relevant references. (Lines 29-30, 52, 57, 59, and 62)
- Tang, H.; Li, P.; Chen, L.; Ma, J.-K.; Guo, H.-H.; Huang, X.-C.; Zhong, R.-M.; Jing, S.-Q.; Jiang, L.-W. The formation mechanisms of key flavor substances in stinky tofu brine based on metabolism of aromatic amino acids. Food Chem. 2022, 392, 133253, doi:10.1016/j.foodchem.2022.133253.
- Skåra, T.; Axelsson, L.; Stefánsson, G.; Ekstrand, B.; Hagen, H. Fermented and ripened fish products in the northern European countries. Journal of Ethnic Foods 2015, 2, 18-24, doi:10.1016/j.jef.2015.02.004.
- Li, H.; Zheng, R.; Zuo, F.; Qian, C.; Yao, Z.; Dong, R.; Zhao, D.; Li, C. Influence of Proteolysis on the Binding Capacity of Flavor Compounds to Myofibrillar Proteins. Foods 2022, 11, 891, doi:10.3390/foods11060891.
- Guo, Z.; Teng, F.; Huang, Z.; Lv, B.; Lv, X.; Babich, O.; Yu, W.; Li, Y.; Wang, Z.; Jiang, L. Effects of material characteristics on the structural characteristics and flavor substances retention of meat analogs. Food Hydrocolloid. 2020, 105, 105752, doi:10.1016/j.foodhyd.2020.105752.
- Zhang, J.; Kang, D.; Zhang, W.; Lorenzo, J.M. Recent advantage of interactions of protein-flavor in foods: Perspective of theoretical models, protein properties and extrinsic factors. Trends Food Sci.Tech. 2021, 111, 405-425, doi:10.1016/j.tifs.2021.02.060.
- Gu, S.; Dai, W.; Chong, Y.; Lyu, F.; Zhou, X.; Ding, Y. The binding of key fishy off-flavor compounds to silver carp proteins: A thermodynamic analysis. RSC advances 2020, 10, 11292-11299, doi:10.1039/d0ra01365j.
